# c-Met Mediated Cytokine Network Promotes Brain Metastasis of Breast Cancer by Remodeling Neutrophil Activities

**DOI:** 10.3390/cancers15092626

**Published:** 2023-05-05

**Authors:** Yin Liu, Margaret R. Smith, Yuezhu Wang, Ralph D’Agostino, Jimmy Ruiz, Thomas Lycan, Gregory L. Kucera, Lance D. Miller, Wencheng Li, Michael D. Chan, Michael Farris, Jing Su, Qianqian Song, Dawen Zhao, Arvind Chandrasekaran, Fei Xing

**Affiliations:** 1Department of Cancer Biology, Wake Forest University School of Medicine, Winston-Salem, NC 27157, USA; 2Department of Biostatistics and Data Science, Wake Forest University School of Medicine, Winston-Salem, NC 27157, USA; 3Department of Hematology and Oncology, Wake Forest University School of Medicine, Winston-Salem, NC 27157, USA; 4Department of Pathology, Wake Forest School of Medicine, Winston-Salem, NC 27157, USA; 5Department of Radiation Oncology, Wake Forest University School of Medicine, Winston-Salem, NC 27157, USA; 6Department of Biostatistics, Indiana University School of Medicine, Indianapolis, IN 47405, USA; 7Bioinspired Microengineering Laboratory (BIOME), Department of Chemical, Biological and Bioengineering, NC A&T State University, Greensboro, NC 27411, USA

**Keywords:** breast cancer, brain metastasis, c-Met, neutrophil, lipocalin 2

## Abstract

**Simple Summary:**

Multiple non-cancerous cells are known to be involved in brain metastasis, and the roles of neutrophils during this event are poorly understood. We aim to understand how tumor-infiltrated neutrophils promote breast cancer brain metastasis and how tumor cells affect the properties of neutrophils. Utilizing patient-based analyses together with our unique animal models, we discovered that several c-Met mediated inflammatory cytokines, including CXCL1/2 and G/GM-CSF, are critical to the neutrophil recruitment and activity in the metastatic lesions. In return, neutrophils activated by those factors secrete a high level of lipocalin 2 (LCN2), which in turn enhances the stemness of tumor cells. Our study revealed novel interactions between neutrophils and brain metastatic cells, which may offer new insight into treating brain metastasis.

**Abstract:**

The brain is one of the most common metastatic sites among breast cancer patients, especially in those who have Her2-positive or triple-negative tumors. The brain microenvironment has been considered immune privileged, and the exact mechanisms of how immune cells in the brain microenvironment contribute to brain metastasis remain elusive. In this study, we found that neutrophils are recruited and influenced by c-Met high brain metastatic cells in the metastatic sites, and depletion of neutrophils significantly suppressed brain metastasis in animal models. Overexpression of c-Met in tumor cells enhances the secretion of a group of cytokines, including CXCL1/2, G-CSF, and GM-CSF, which play critical roles in neutrophil attraction, granulopoiesis, and homeostasis. Meanwhile, our transcriptomic analysis demonstrated that conditioned media from c-Met high cells significantly induced the secretion of lipocalin 2 (LCN2) from neutrophils, which in turn promotes the self-renewal of cancer stem cells. Our study unveiled the molecular and pathogenic mechanisms of how crosstalk between innate immune cells and tumor cells facilitates tumor progression in the brain, which provides novel therapeutic targets for treating brain metastasis.

## 1. Introduction

About 30% of patients with metastatic breast cancer eventually develop brain metastasis, which significantly affects the cognitive and sensory function as well as morbidity of the patients [1,2]. Because of the low delivery efficiency of most anti-cancer drugs due to the blockage of the blood-brain barrier (BBB), tumor cells find the brain as a sanctuary that provides them with sufficient nutrients for them to survive and thrive. Based on the observations from patients and animal models, metastatic tumor cells disseminate in the brain at different stages of cancer progression, and some cells eventually outgrow when they adapt themselves to the foreign environment [3]. Multiple tyrosine kinase receptor pathways have been shown to be involved in brain metastasis, and several tyrosine kinase inhibitors (TKIs) showed promising results in treating brain metastasis both in pre-clinical models and patients [4,5,6]. In our previous study, we found that brain metastatic cancer cells highly express c-Met, a tyrosine kinase receptor that regulates cell migration, invasion, and survival in breast cancer by activating downstream signaling, including MAPK, STAT, PI3K, and NFκB pathways [7]. We also found that upregulation of c-Met in brain metastatic cells creates a highly inflammatory environment in the brain through various cytokines and chemokines, including CXCL1, IL1β, and IL8. Such inflammatory environment activates astrocytes and endothelial cells, which eventually create a favorable microenvironment for the cancer cells.

The normal brain was considered an immune-privileged organ with a limited presence of immune cells [8]. However, recent studies have shown that large amounts of immune cells, including T cells, B cells, macrophages, and neutrophils, are recruited to brain metastatic sites due to the leakiness of blood vessels and chemotactic factors secreted from tumor cells [9,10]. Neutrophils are the most abundant innate immune cells in the circulation, representing 50–70% of all leukocytes [11]. The primary role of neutrophils is to act as the first responder after infection, which kills microorganisms by releasing granule proteins [12]. In cancer settings, neutrophils are reported to have both anti- and pro-tumoral roles depending on the stage and site of the tumor [11]. Neutrophils have been shown to suppress tumor progression by releasing tumor-killing nitric oxide, reactive oxygen species (ROS), and hydrogen peroxide [13,14,15]. On the other hand, neutrophils are known to promote tumor progression in various ways, including enhancing the survival of circulating tumor cells (CTC) in the circulation, facilitating CTC extravasation, promoting angiogenesis, and creating an immune suppressive microenvironment by suppressing T cell functions [16]. However, the roles of neutrophils in brain metastasis are still unclear. Several studies reported that a high ratio of neutrophils to lymphocytes in brain metastasis patients is associated with decreased survival [17,18,19]. Zhang et al. have shown that neutrophils are immune suppressive cells in brain lesions, and targeting those neutrophils might be a potential therapeutic strategy for treating brain metastasis [10]. Like macrophages, heterogeneous populations of neutrophils known as N1 and N2 neutrophils have been identified in the tumor microenvironments based on their surface markers and cytokine profiles [15]. We have demonstrated that LCN2, also known as neutrophil gelatinase-associated lipocalin (NGAL), is highly upregulated in N2 neutrophils. LCN2 is a granule protein that plays a critical role in the innate immune response to bacterial infection by iron sequestration [20]. Interestingly, LCN2 levels in serum and cerebrospinal fluid have been suggested as a potential marker for mild cognitive impairment and Alzheimer’s disease, suggesting a role of LCN2-mediated iron homeostasis in the brain [21,22]. In breast cancer, LCN2 has been reported to promote invasiveness and cancer stemness by enhancing iron uptake [23,24].

Our findings have elucidated how c-Met mediated inflammatory cytokines affect neutrophil functions during different stages of breast cancer brain metastasis and how LCN2 from N2 neutrophils promote tumor progression, which provides insights into developing potential therapeutic strategies by targeting c-Met signaling and N2 neutrophils.

## 2. Material and Methods

### 2.1. Human Samples

Formalin-fixed paraffin-embedded tumor tissues from breast cancer brain metastatic patients (*n* = 26) were retrieved from the tumor tissue bank of Wake Forest Baptist Comprehensive Cancer Center in accordance with ethical guidelines. All samples were collected under the Wake Forest School of Medicine IRB (Institute Review Board) approved protocol IRB00060939.

### 2.2. Cell Culture and Reagents

Human breast carcinoma cell lines MCF7, SKBR3, and MDA-MB-453 were purchased from American Type Culture Collection. MDA-MB231BrM2a (231BrM) was a kind gift from Dr. Massague (Memorial Sloan-Kettering Cancer Center). SKBrM3 and 231BrM are brain metastatic cell lines derived from parental SKBR3 and MDA-MB-231 cells through several rounds of in vivo selections. Human brain microvascular endothelial cells (HBMEC) were purchased from Lonza. Mouse neutrophils were isolated from C57B6 mice bone marrow with a neutrophil isolation kit (Miltenyi Biotec, Gaithersburg, MD, USA, #130-097-658).

### 2.3. Tissue Culture Conditions

SKBrM3, MCF7, and 231BrM were cultured in DMEM medium supplemented with 10% FBS. MDA453 was cultured in RPMI medium supplemented with 10% FBS. HBMECs were maintained in an AGM medium supplemented with EGM-Plus and EGM-2 growth media (Lonza, Allendale, NJ, USA). Mouse neutrophils were cultured in RPMI medium with 10% FBS. Cells were grown at 37 °C in a 5% CO_2_ atmosphere.

### 2.4. Plasmids and Reagents

The doxycycline-inducible sh-cMet cell line and MET overexpression cell lines were generated as previously described [7]. Briefly, the doxycycline-inducible shMet plasmid was generated by cloning sh-c-Met sequence (annealed two oligonucleotides; 5′-CCGGCAGAATGTCATTCTACATGAGCTCGAGCTCATGTAGAATGACATTCTGTTTTT-3′, and 5′-AATTAAAAACAGAATGTCATTCTACATGAGCTCGAGCTCATGTAGAATC ACATTCTG-3′) into Tet-PLKO-Puro vector (Addgene). The c-Met overexpression plasmid was purchased from Origene. Recombinant LCN2 and G-CSF were purchased from R&D system (#1857-LC and #214-CS). LCN2 antibody was purchased from R&D system (#AF1857). Doxycycline was purchased from Sigma Co, Burlington, NJ, USA.

### 2.5. Conditioned Medium (CM) Preparation

For the preparation of the 1st CM from tumor cells, 231BrM-tet-shMet and SKBrM3-tet-shMet cells were cultured in medium with doxycycline (1 µg/mL) prior to CM preparation. Forty-eight hours later, the Dox−containing medium was replaced with serum-free DMEM. Thirty-six hours later, CM was collected, and cell debris was removed by centrifugation. For secondary neutrophil CM, 1 million neutrophils were seeded in cancer cell CM for 36 h. The medium was then collected, and cell debris was removed by centrifugation.

### 2.6. Neutrophil Isolation

Mouse neutrophils were isolated from the bone marrow of 8 to 12-week-old C57/B6 mice with a mouse neutrophil isolation kit (Miltenyi Biotec, Gaithersburg, MD, USA, #130097658) following the manufacturer’s instruction. Briefly, the bone marrow of the mouse tibia was harvested and suspended in PBS. Cells were labeled with various biotin-labeled negative selection antibodies and then labeled with anti-biotin microbeads. Unlabeled neutrophils were collected from the flow-through.

### 2.7. Immunohistochemistry

Human breast cancer brain metastasis samples were obtained from CHTN. Samples were sectioned with 10 µm thickness from the formaldehyde-fixed and paraffin-embedded tissue specimens. The sections were deparaffinized, and antigens were retrieved by heating the slides in 10 mM sodium citrate (pH 6.0) at 85 °C for 30 min. The slides were treated with 3% H_2_O_2_ and then incubated overnight at 4 °C with ELA2 antibody (1/100; Abcam #ab68672). The sections were then incubated with secondary antibodies and visualized using the Envision-plus kit (Agilent Dako, Santa Clara, CA, USA).

### 2.8. ELISA Assays

Cancer cells were cultured with serum-free medium for 36 h, and conditioned medium was collected, and they were subjected to ELISA assays for CXCL1, CXCL2, G-CSF, and GM-CSF (R&D system, Minneapolis, MN, USA #DY275, #DY276-05, #DY214, and #DY215, respectively). Neutrophils were cultured in a cancer cell-conditioned medium, or RPMI, for 36 h, and the secondary conditioned medium was collected and subjected to ELISA assays for LCN2 (Abcam, Waltham, MA, USA #ab199083).

### 2.9. Quantitative Real-Time PCR

Total RNA was isolated from the cells and reverse transcribed as described previously. The cDNA was then amplified with a pair of forward and reverse primers (Appendix A). PCR reactions were performed using CFX Connect (Bio-Rad, Hercules, CA, USA) and iTaq Universal SYBR Green Supermix (Bio-Rad). The thermal cycling conditions were composed of an initial denaturation step at 95 °C for 5 min, followed by 40 cycles of PCR using the following profile: 94 °C, 30 s and 60 °C, 30 s.

### 2.10. Animal Experiments

All animal experiments were done in accordance with protocols approved by the Wake Forest Institutional Animal Care and Use Committee. For intracardiac injection, 2 × 10^5^ luciferase-labeled cancer cells in 100 µL PBS were injected into the left ventricle of nude mice (7–8 weeks). For neutrophil depletion, 2 doses of 200 µg of IgG (BioXcell, Lebanon, PA, USA #BE0085) or Ly6G (BioXCell, Lebanon, PA, USA #BE0075-1) were injected into each mouse before cancer cell inoculation every three days and maintained injection twice a week after cancer cell injection. 

For imaging, mice were immediately measured after intracardiac injection to confirm successful injection using an IVIS Xenogen bioimager (Caliper). The brain metastasis progression was monitored twice a week, and the luminescence was quantified at the indicated time points. At the endpoint of the study, the whole brain was removed and incubated in RPMI-1640 medium with 0.6 mg/mL luciferin for 5 min, and photon flux was measured by an IVIS Xenogen bioimager. 

### 2.11. Flow Cytometry

For Ly6G and CD11b, neutrophil single-cell suspensions were stained with anti-mouse CD11b-PECy5 (1:1000, BioLegend, San Diego, CA, USA # 103012), and Ly6G-APC (1:1000, eBioscience, San Diego, CA, USA # 17-9668-82). Cell debris and dead cells were excluded from the analysis based on scatter signals. For AnnexinV and 7AAD, the neutrophil single-cell suspension was stained with PE Annexin V Apoptosis detection kit (BD Bioscience, Franklin Lakes, NJ, USA #559763). Data were collected on an Accuri C6 analyzer and analyzed using Flowjo.

### 2.12. Neutrophil Migration Assay

Twenty thousand freshly isolated neutrophils were labeled with cell tracker dye (Invitrogen, Waltham, MA, USA #C2925) and added to the upper chamber (BD Bioscience, Franklin Lakes, NJ, USA, #363096). Cancer cell-conditioned medium was added to the lower chamber. Cells migrated to the lower chamber were imaged and counted after 2 h.

### 2.13. Mammary Sphere Formation Assay

One thousand cells were plated in an ultra-low attachment plate (Corning, Corning, MI, USA #3473) in a mammosphere-forming medium containing DMEM/F12 supplemented with 2% B27 (Invitrogen), 20 ng/mL EGF, and 4 µg/mL Insulin (Sigma Co, Burlington, VT, USA). Spheres were counted, and images were taken on day 7.

### 2.14. Tube Formation Assay

Tube formation assay was performed as previously described [7]. Briefly, the 96-well plate was coated with 50-μL growth factor-reduced Matrigel (BD Bioscience, Franklin Lakes, NJ, USA) until it was solidified. HBMECs cells were suspended in a serum-free conditioned medium and seeded on top of the growth factor-reduced Matrigel. After 6 h, photos were taken under a microscope, and the number of tube structures per field was counted.

### 2.15. RNA Sequencing

RNAs were extracted from cancer cells or neutrophils using RNeasy Micro Kit (Qiagen, Germantown, USA Cat #74004), and the concentration was measured with a Nano drop (Thermo Fisher Scientific Waltham, MA, USA). Then, 75bp-paired-end RNA sequencing was performed using the Illumina NextSeq 500 system with a total of 50 M reads at the Wake Forest Cancer Genomics Share Resource. Gene expression was analyzed using the R package DESeq2. For the volcano plot, the adjusted *p*-value of less than 0.05 and FDR of less than 0.05 were applied for statistical significance.

### 2.16. Cohort Analysis

For cancer cohort analysis, we compiled a microarray dataset of 710 patients from GEO (accession numbers: GSE12276, GSE2034, GSE2603, GSE5327, and GSE14020). These datasets were all normalized using MAS5.0, and each microarray was centered to the median of all probes. For each patient, brain-metastasis-free survival was defined as the time interval between the surgery and the diagnosis of metastasis.

### 2.17. Statistical Analysis

All analyses were calculated by GraphPad Prism if not specified. Data are presented as mean ± sd. The *p*-value is calculated by unpaired Student’s *t*-test if not specified. Significance between each group was represented as * *p*  <  0.05, ** *p*  <  0.01, or *** *p* < 0.001. Principal analysis of RNA sequencing data was conducted by pcaExplorer [25].

## 3. Results

### 3.1. c-Met Expression in Tumor Cells Is Associated with the Enrichment of Neutrophils in Brain Metastatic Sites

In our previous study, we generated two tetracycline-inducible c-Met knockdown cells from 231BrM and SKBrM3, two brain tropical cell lines, and found that knockdown of c-Met in 231BrM-Tet-shMet cells by feeding mice with doxycycline water significantly suppressed brain metastasis in vivo [7]. In this study, we validated our previous findings by using the SKBrM3-Tet-shMet cells (Figure 1A,B). To further investigate how the c-Met pathway regulates brain metastasis, we performed Ingenuity Pathway Analysis (IPA) by analyzing the top 200 c-Met signature genes generated from 231BrM-Tet-shMet cells and SKBrM3-Tet-shMet cells with or without knockdown of c-Met (Figure 1C). Interestingly, we found that TREM1 and GM-CSF signaling pathways which are known to play pivotal roles in neutrophil maturation and survival, were ranked as one of the most enriched pathways [11]. In addition, signaling pathways involved in granulocyte adhesion and crosstalk between innate and adaptive immune cells were also identified (Figure 1D). These data strongly suggest that activation of c-Met signaling in brain metastatic tumor cells is associated with an atypical neutrophil activity which may influence tumor growth in the brain. To further validate our findings, we stained brain metastatic samples from mice inoculated with 231BrM-Tet-shMet and SKBrM3-Tet-shMet using neutrophil elastase (ELA2) antibody, a highly specific neutrophil marker. As shown in Figure 1E, the knockdown of c-Met significantly reduced the amount of infiltrated neutrophils in the metastatic lesions. We also stained clinical samples (*n* = 26) derived from brain metastatic lesions with ELA2 antibody and observed the presence of neutrophils in ~80% of brain metastatic samples with different patterns (Figure 1F). We also found that most of the neutrophils were presented in the peritumoral area, followed by infiltrating and extravasation phenotypes. Under certain conditions, neutrophils respond to pro-inflammatory stimuli by forming web-like structures composed of histones and decondensed chromatin, collectively termed Neutrophil Extracellular Traps (NETs). Growing evidence has shown that NETs promote tumor growth and metastasis in various types of cancers [26,27]. Notably, NETs were found in two samples, suggesting a potential role of NETs in promoting brain metastasis. Based on these observations, it is clear that neutrophils are presented in the brain metastatic lesions, and their activities are associated with c-Met expression in tumor cells.

### 3.2. c-Met Pathway Promotes the Release of a Group of Neutrophil-Related Cytokines 

Neutrophils are differentiated from granulocyte–monocyte progenitors in the bone marrow under the direction of G-CSF or GM-CSF and then released to the circulation followed by CXCL1/2/5/8 mediated signaling [11]. We examined the expression of these factors in our RNA sequencing data and found that the knockdown of c-Met significantly downregulated the expression of G-CSF, GM-CSF, CXCL1, and CXCL2 but not CXCL5 (Figure 2A). In a combined cohort of breast cancer patients, c-Met expression is positively correlated to G-CSF, GM-CSF, CXCL1, and CXCL2 (Appendix A). We also found that high expression of G-CSF and CXCL1 but not GM-CSF or CXCL2, in primary tumors is significantly correlated with poor brain metastasis-free survival (Figure 2B and Appendix A). Since communications between neutrophils and cancer cells are mainly through systemic effect mediated by cytokines, we prepared conditioned medium (CM) from cancer cells with (Dox+) or without (Dox−) knockdown of c-Met followed by measuring cytokine concentration by ELISA (Figure 2C,D). Indeed, the knockdown of c-Met in two brain metastatic cells significantly reduced the secretion of G-CSF, GM-CSF, CXCL1, and CXCL2 (Figure 2D). We also found that overexpression of c-Met in MCF7 and MDA-MB-453 significantly increased G-CSF and CXCL1 but not GM-CSF concentration in CM, while the amount of CXCL2 was not detectable (Figure 2E and Appendix A).

### 3.3. CM of Brain Metastatic Cells Regulates Neutrophil Activities in a c-Met Dependent Manner

We have demonstrated that upregulated c-Met signaling in tumor cells enhanced the expression of a group of neutrophil-related cytokines. Therefore, we decided to investigate whether CM from cells with or without knockdown of c-Met modulates neutrophil behavior. When granulocyte–monocyte progenitors in the bone marrow are stimulated by G-CSF or GM-CSF, they will successively differentiate into myoblasts, promyelocytes, myelocytes, metamyelocytes, band cells, and eventually neutrophils [11]. During the differentiation, neutrophils change their shape of nuclear with the increased expression of LY6G and CD11b on the surface [11]. Neutrophils in different maturation stages are known to have distinctive roles during cancer progression [28]. First, we will examine whether CM from brain metastatic cells with different expressions of c-Met affect neutrophil differentiation and maturation. Neutrophils were isolated from the bone marrow of C57/B6 mice, and the majority (~80%) of the isolated neutrophils were considered immature (Appendix A). Neutrophils were treated with CM from brain metastatic cells with or without knockdown of c-Met, followed by analyzing their nuclear morphology and expression of surface markers. We found that Dox− CM derived from 231BrM-Tet-shMet cells significantly promoted the maturation of mouse neutrophils compared to RPMI and Dox+ CM (Appendix A). Meanwhile, treatment of Dox− CM from 231BrM-Tet-shMet and SKBrM3-Tet-shMet cells significantly increased LY6G^hi^/CD11b^hi^ population in mouse neutrophils and knockdown of c-Met attenuated this effect, suggesting c-Met pathway-mediated soluble factors from tumor cells are essential for the maturation of neutrophils (Figure 3A). Neutrophils are retained in the bone marrow until mobilized by CXCL1/2/5/8 stimulation in the circulation [11]. In addition, CXCL1/2/5/8 are also responsible for the recruitment of circulating neutrophils to the tumor [11]. Next, we performed a neutrophil migration assay using the trans-well system to examine whether activation of c-Met signaling in tumor cells is able to attract neutrophils to the metastatic sites. Indeed, we observed a significantly higher amount of migrated neutrophils in the Dox− CM treated group compared to the RPMI and Dox+ CM treated groups (Figure 3B). After entering the cancer microenvironment, whether neutrophils are able to influence cancer progression is highly dependent on how long they can survive, as neutrophils are known to have a very short lifespan. However, neutrophils are reported to have prolonged life in tumors, and several factors, including G-CSF and GM-CSF in the tumor microenvironment, have been shown to promote their survival [11]. To test whether c-Met expression in cancer cells affects neutrophil survival, we performed AnnexinV/7AAD staining in neutrophils treated with tumor CM with or without knockdown of c-Met. As shown in Figure 3C, Dox− CM treatment significantly prolonged the survival of neutrophils compared to RPMI and Dox+ CM treatment. Our data suggest that the c-Met pathway in tumor cells regulates a group of cytokines that are indispensable to neutrophil maturation, attraction, and survival.

### 3.4. Depletion of Neutrophils Suppresses Brain Metastasis In Vivo

To examine whether neutrophils contribute to brain metastasis in vivo, we depleted neutrophils by Ly6G antibody in nude mice one week prior to the inoculation of brain metastatic cancer cells (Figure 4A). Successful depletion of neutrophils in the circulation was confirmed by Ly6G/CD11b FACS (Figure 4B,C). Indeed, the depletion of neutrophils significantly prolonged the survival of mice inoculated with SKBrM3 (Figure 4D,E) and 231BrM (Figure 4F,G) cells with decreased tumor burden in the brain. Several groups reported that neutrophils promote cancer metastasis by facilitating cancer extravasation, enhancing the survival of circulating tumor cells, and inducing angiogenesis. Interestingly, the depletion of neutrophils also significantly reduced the bone metastasis burden, suggesting a general role of neutrophils during tumor metastasis.

### 3.5. Phenotypic Switch of Neutrophils Is Associated with c-Met Signaling in Tumor Cells

Studies have shown that neutrophils can either have suppressive (N1) or supportive (N2) roles in tumor progression, depending on the populations identified in the tumor microenvironment [10,15]. CCL2, ARG1, ARG2, NOS2, and CCL5 are well-known N2-associated cytokines that promote tumor growth, invasion, and immune suppression. We found CM from brain metastatic cells significantly upregulated expression of N2 cytokines in neutrophils compared to RPMI medium, and knockdown of c-Met in cancer cells attenuated this effect (Figure 5A,B). To gain a deeper understanding of the transcriptomic remolding in those neutrophils and whether such remolding depends on c-Met signaling in tumor cells, we performed RNA sequencing in mouse neutrophils treated with CM of 231BrM-tet-shMet cells with or without knockdown of c-Met. Among the 39,175 genes identified, 11,868 genes showed significant differential expression between DOX− CM and DOX+ CM treated neutrophils (Appendix A, Appendix A). Principle analysis also demonstrated unique transcriptome profiles between cells treated with different CM (Figure 5C). Neutrophils mainly interact with other cells through external proteins and cytokines [20]. Therefore, we examined the expression of neutrophils’ external proteins, including granule proteins, netome, cytonme, and extracellular vesicles containing proteins from neutrophils [20]. Surprisingly, we observed overall downregulation of these genes except lipocalin 2 (LCN2) and catalase (CAT) in DOX− CM treated neutrophils compared to DOX+ CM treated or not treated neutrophils (Figure 5D). Among the 39 neutrophil-secreted cytokines, 18 showed differential expression between DOX− CM and DOX+ CM treated neutrophils (Appendix A). Our data indicate that the release of c-Met-dependent cytokines from brain metastatic cells reprograms the neutrophils towards the N2 phenotype through key genes that are associated with neutrophil activities and homeostasis.

### 3.6. Neutrophils Promote Cancer Cell Stemness through LCN2

To examine how N2 neutrophils promote brain metastasis, we selected three genes from the RNA sequencing data, including LCN2, CAT, and PROK2 (BV8), that showed significant upregulation in DOX− CM compared to DOX+ and not treated cells. LCN2 and BV8 but not CAT expression, were validated by qRT-PCR (Figure 6A and Appendix A). BV8 secreted from neutrophils has been reported to promote angiogenesis by activating endothelial cell proliferation [29,30]. However, the tube formation ability of mouse brain endothelial cells was not affected by the treatment of DOX− 2nd or DOX+ 2nd CM from neutrophils (Figure 6B and Appendix A). This might be due to the DOX+ 1st CM-treated neutrophils having high expression of VEGFA, which rescued the pro-angiogenic effect of DOX+ 2nd CM (Appendix A). Because most of the brain metastatic cells are highly invasive, the co-culture of neutrophils failed to further enhance the migration ability of brain metastatic cells (Appendix A). Recent studies have shown that extracellular LCN2 enhances the stemness of cancer cells by elevating iron uptake [23,24]. We first confirmed the secretion of LCN2 from neutrophils by ELISA and found that DOX− CM significantly increased the amount of LCN2 secretion by neutrophils compared to DOX+ CM treatment (Figure 6C). Next, we performed a mammary sphere formation assay to examine whether neutrophils promote cancer stemness by LCN2. First, we pretreated cancer cells with DMEM, DOX− 2nd CM, DOX+ 2nd CM, and DOX+ 2nd CM with recombinant LCN2, followed by mammosphere formation. Indeed, mammosphere formation of tumor cells was significantly upregulated by the DOX− 2nd CM compared to DOX+ 2nd CM-treated cells, and recombinant LCN2 rescued mammosphere formation in DOX+ 2nd CM-treated cells (Figure 6D). Next, we pretreated cancer cells with a gradient concentration of LCN2 and found that LCN2 promotes cancer cell sphere formation in a concentration-dependent manner (Figure 6E). We also found that the addition of LCN2 neutralizing antibodies significantly reduced the mammary sphere formation ability of DOX− 2nd CM treated SKBrM3 cells. These data suggest that neutrophil-derived LCN2 promotes the stemness of brain metastatic cell lines. To identify which c-Met-regulated cytokines regulate LCN2 expression in neutrophils, we used the Cytosig database, which reveals cellular response to specific signaling molecules obtained from publicly available data [31]. The results showed that G-CSF is the major cytokine that upregulates LCN2 expression (Figure 6G). To validate our findings, we treated neutrophils with G-CSF followed by LCN2 ELISA and found an increased amount of LCN2 in the CM upon G-CSF stimulation (Figure 6H). These data suggest that c-Met-mediated G-CSF secretion in cancer cells elevates LCN2 expression in neutrophils.

In summary, we have demonstrated that elevated c-Met pathway in brain metastatic cells enhances the expression of several cytokines, including G-CSF, GM-CSF, CXCL1, and CXCL2, which regulates neutrophils maturation, attraction, survival, and polarization. We also found that N2 neutrophils promote brain metastasis by supporting cancer stemness through LCN2. 

## 4. Discussion

The c-Met pathway has been known to play important roles in cell survival, migration, and invasion [32]. In our studies, we have shown that the c-Met pathway in brain metastatic cells regulates the expression of a wide variety of cytokines which contribute to the attraction, maturation, and survival of neutrophils in the tumor microenvironment. We also found that N2 neutrophils promote brain metastasis by secreting G-CSF-mediated LCN2, which enhances the cancer stemness of brain metastatic cells. Both metastasis-promoting and suppressing roles of neutrophils have been reported. Several groups have reported that neutrophils promote breast cancer metastasis by forming NETs that attract cancer cells to the metastatic sites [26,27]. Casbon et al. have reported that neutrophils reprogrammed by tumor cell-derived G-CSF are immunosuppressive [33]. On the other hand, Li et al. demonstrated that tumor cell-derived G-CSF could either be pro-metastatic or anti-metastatic depending on the presence or absence of host NK cells, and the lack of NK cells, but not T cells or B cells, significantly attenuated the pro-metastatic ability of G-CSF-stimulated neutrophils [34]. They also found ROS released from neutrophils kills both tumor and NK cells, and targeting the ROS pathway in neutrophils abrogates the pro-metastatic and anti-metastatic effects of neutrophils in vivo. Another study showed that H_2_O_2_ secreted from tumor-entrained neutrophils (TENs) suppresses tumor growth in the lung, and depletion of neutrophils significantly promoted lung metastasis in the 4T1 spontaneous model [35]. They also showed that G-CSF-stimulated neutrophils failed to suppress lung metastasis, while tumor-derived CCL2 promotes the cytotoxic effect of TENs. In our study, we have demonstrated that G-CSF is highly expressed in c-Met high cells and patients with brain metastasis. We also found that LCN2, a key stemness factor, is induced in G-CSF-stimulated neutrophils. Therefore, multiple factors may affect the function of neutrophils during the tumor progression, including the stage and site of the tumor, properties of tumor cells, and different immune components in the tumor microenvironment. 

Most of the previous neutrophil-related studies focused on how neutrophil external proteins and cytokines contribute to tumor progression [20]. However, additional tumor-promoting mechanisms of neutrophils have been revealed. Wculek et al. reported that neutrophils promote breast cancer lung metastasis by enhancing cancer stemness through neutrophil-derived leukotrienes produced by the leukotriene-generating enzyme arachidonate 5-lipoxygenase (Alox5) [36]. However, upregulation of Alox5 in neutrophils treated with c-Met high CM was not significant in our RNA sequencing data, suggesting that neutrophils are able to promote cancer stemness through multiple pathways. It has been reported that lung mesenchymal cells in the pre-metastatic niche induce an increase of lipids storage in neutrophils through the PGE2- HIF1α–HILPDA axis, and those neutrophils promote the growth of disseminated cancer cells by feeding them lipids during the early stage of colonization [37]. Interestingly, HILPDA is one of the top upregulated genes in neutrophils treated by c-Met high CM, suggesting that c-Met high cells might also stimulate neutrophils to provide lipids for cancer cells in the brain (Appendix A). In conclusion, the c-Met pathway plays an important role in neutrophil regulation, which might not be fully discovered in this study and deserves further investigation.

We observed an overall decrease of external and granule proteins-related genes, an indicator of low-density neutrophils (LDNs) in neutrophils treated with c-Met high CM compared to c-Met low CM [38,39]. On the other hand, we found that c-Met high CM significantly increased the percentage of mature neutrophils, which are usually recognized as high-density neutrophils. Therefore, a subpopulation of mature neutrophils with low density is likely to be specifically enriched upon exposure to c-Met high CM. Notably, mature LDNs, which are frequently found in PBMC from cancer patients with poor prognosis, are known as immunosuppressive and capable of forming NETs [38,40,41]. However, the origin and exact cytokines that induce the differentiation and degranulation of mature LDNs are still unclear. Our study indicates the tumoral c-Met pathway potentially drives the formation of mature LDNs.

Capmatinib and Tepotinib are two FDA-approved Tyrosine Kinase Inhibitors (TKIs) that are used to treat non-small cell lung cancer (NSCLC) patients with MET exon 14 skipping [42,43]. Notably, Capmatinib and Tepotinib are able to cross the blood-brain barrier [44,45]. Data from GEOMETRY mono-1 study showed that 12 out of 13 patients with brain metastases received a benefit from the treatment, including four patients who had a complete remission of the tumor after Capmatinib treatment [46]. Data from the VISION study showed that 13 out of 15 patients with brain metastases achieved intracranial disease control, including three patients who had complete remission after Tepotinib treatment [47]. However, MET mutation and amplification are rare events (around 1%) in breast cancer patients, while MET protein is commonly overexpressed in breast cancer tissues and brain metastatic samples [32]. Therefore, it is reasonable to speculate that breast cancer patients with brain metastasis may also receive clinical benefits from c-Met inhibitors. Interestingly, the c-Met pathway in neutrophils has been shown to be critical for neutrophil mobilization and recruitment [13,48], suggesting that inhibition of c-Met may have dual effects by targeting both tumor cells and neutrophils.

## 5. Conclusions

In summary, we discovered that neutrophils are excessively presented in the brain metastatic lesions in both human and animal tissue samples. Depletion of neutrophils in mice significantly reduced the brain metastatic burden in vivo. Our previous and current data also demonstrated that c-Met signaling plays a pivotal role during brain metastasis, and overexpression of c-Met in brain metastatic cells promotes the secretion of various inflammatory cytokines, including CXCL1/2 and G/GM-CSF, which enhance the neutrophil infiltration and survival. Our RNA-seq analysis also identified LCN2 as one of the major factors that are secreted by neutrophils upon the stimulation of the c-Met high tumor cells. Lastly, LCN2 released from neutrophils significantly enhanced the stemness of tumor cells.

## Figures and Tables

**Figure 1 cancers-15-02626-f001:**
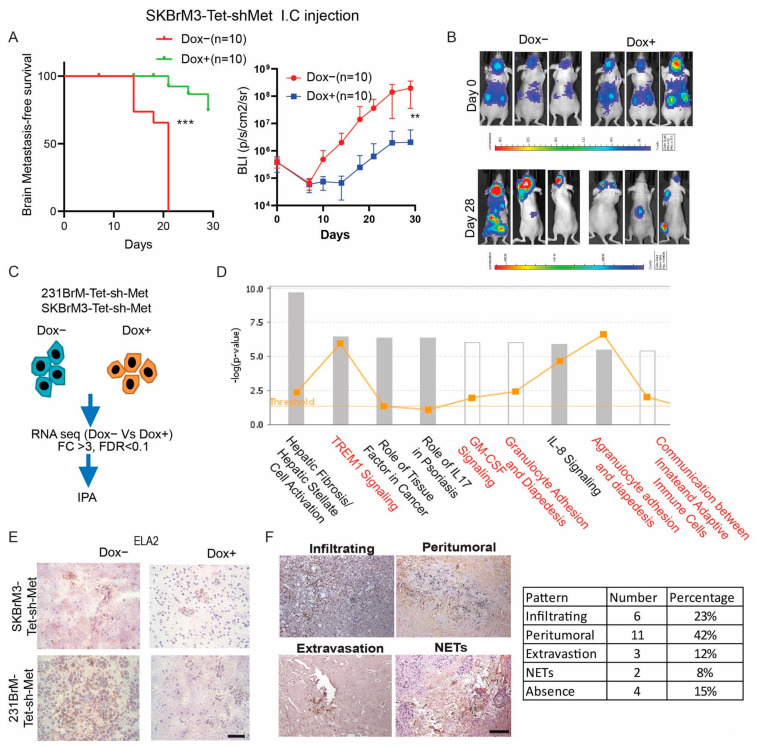
c-Met signaling in cancer cells is associated with neutrophil activity in brain metastasis (**A**) Brain metastasis-free survival and brain bioluminescence of SKBrM3-Tet-shMet cells with or without doxycycline treatment after tumor inoculation. Doxycycline was given in drinking water immediately after injection. I.C., intracardiac. (**B**) Representative picture of mice at day 0 and day 28 after tumor inoculation. (**C**) Flowchart of pathway analysis. (**D**) A list of 200 top-upregulated genes was inputted into QIAGEN Ingenuity Pathway Analysis software for canonical pathway enrichment analysis. Pathways enriched in 231BrM-Tet-shMet and SKBrM3-Tet-shMet without doxycycline compared to cells treated with doxycycline (1 µg/mL). (**E**) Representative picture of ELA2 staining in brain metastasis from mice. Scale bar: 20 µm (**F**) Left, ELA2 staining of brain metastasis sample from breast cancer patients. Scale bar: 100 µm. Right, quantification of neutrophil distribution pattern (** *p*  <  0.01, or *** *p* < 0.001).

**Figure 2 cancers-15-02626-f002:**
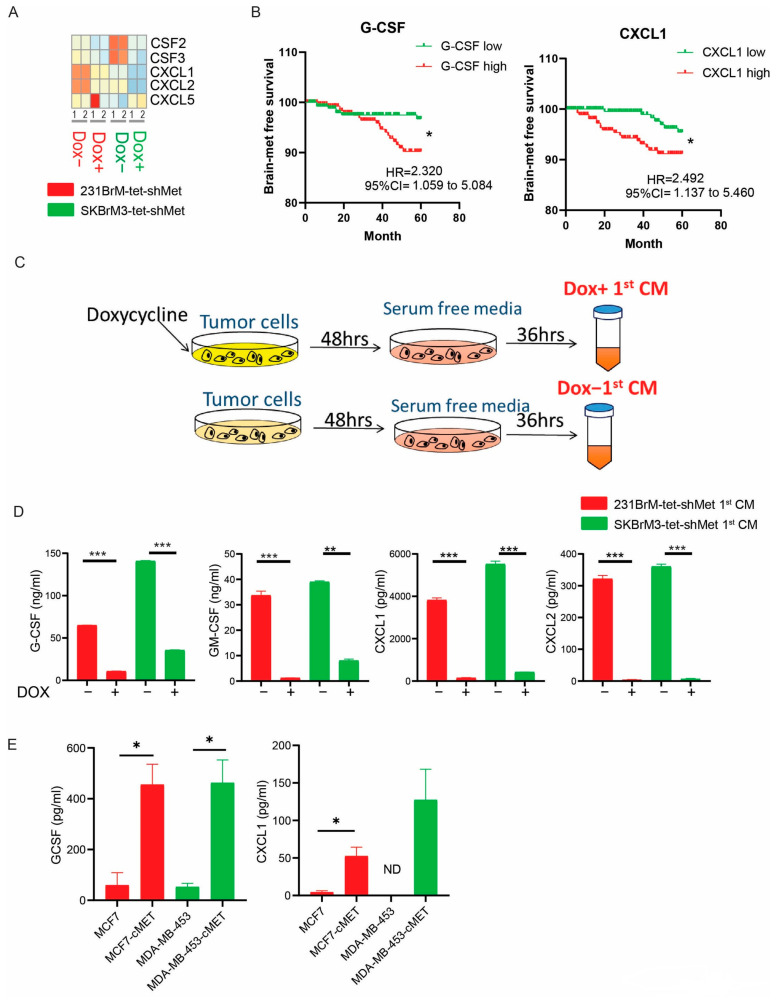
c-Met signaling regulates expression of key neutrophil granulopoiesis and homeostasis factors (**A**) Heatmap of G-CSF (CSF3), GM-CSF (CSF2), and CXCL1/2/5 expression in cells with or without doxycycline treatment from RNA sequencing data. (**B**) Brain metastasis-free survival analyses of G-CSF and CXCL1 in 710 breast cancer patients using the combined GEO databases (GSE12276, GSE2034, GSE2603, GSE5327, and GSE14020). Patients were divided into two groups based on the expression status of the gene in their primary tumors. (**C**) Scheme of conditioned medium (CM) preparation. Notably, there is no doxycycline in DOX+ CM. Cells were pretreated with doxycycline for 48 h before CM preparation. (**D**) ELISA of CXCL1/2, GCSF, and GM-CSF in 231BrM-Tet-shMet and SKBrM3-Tet-shMet CM. (**E**) ELISA of CXCL1 and GCSF in MCF7, MCF7-cMet, MDA-MB-453 and MDA-MB-453-cMet CM (* *p*  <  0.05, ** *p*  <  0.01, or *** *p* < 0.001).

**Figure 3 cancers-15-02626-f003:**
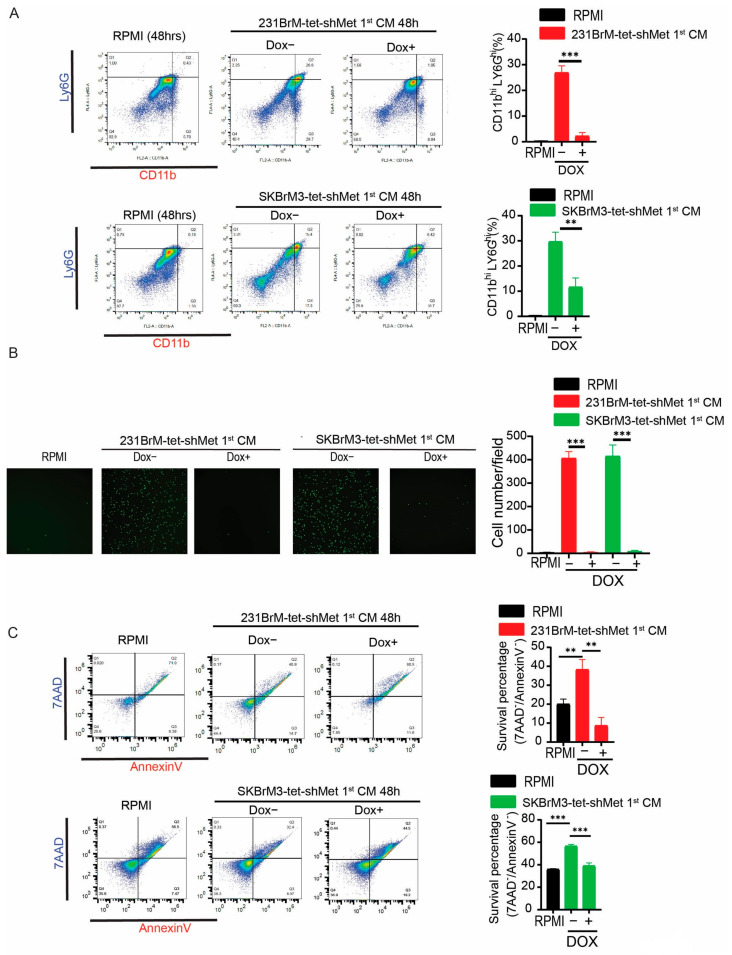
c-Met mediated soluble factors regulate neutrophil mobilization, maturation, and lifespan (**A**) Neutrophils isolated from mouse bone marrow were treated by indicated CM or RPMI for 48 h. Neutrophils were then stained by CD11b and Ly6G antibodies, followed by flow cytometry analysis. Left, representative flow cytometry graph. Right, quantification of CD11b^hi^/Ly6G^hi^ percentage. (**B**) Neutrophils were seeded in the upper chamber of a trans-well with indicated 1st CM or RPMI in the lower chamber. Number of neutrophils that migrated to the lower chamber was counted after 3 h. Left, representative picture at the endpoint. Right, quantification of cells migrated to the lower chamber. (**C**) Neutrophils isolated from mouse bone marrow were treated by CM or RPMI for 48 h. Neutrophils were stained by 7AAD and AnnexinV and followed by flow cytometry analysis. Left, representative flow cytometry graph. Live cells that were not undergoing apoptosis were gated in the lower left panel. Right, quantification of survived neutrophils after 48 h (** *p*  <  0.01, or *** *p* < 0.001).

**Figure 4 cancers-15-02626-f004:**
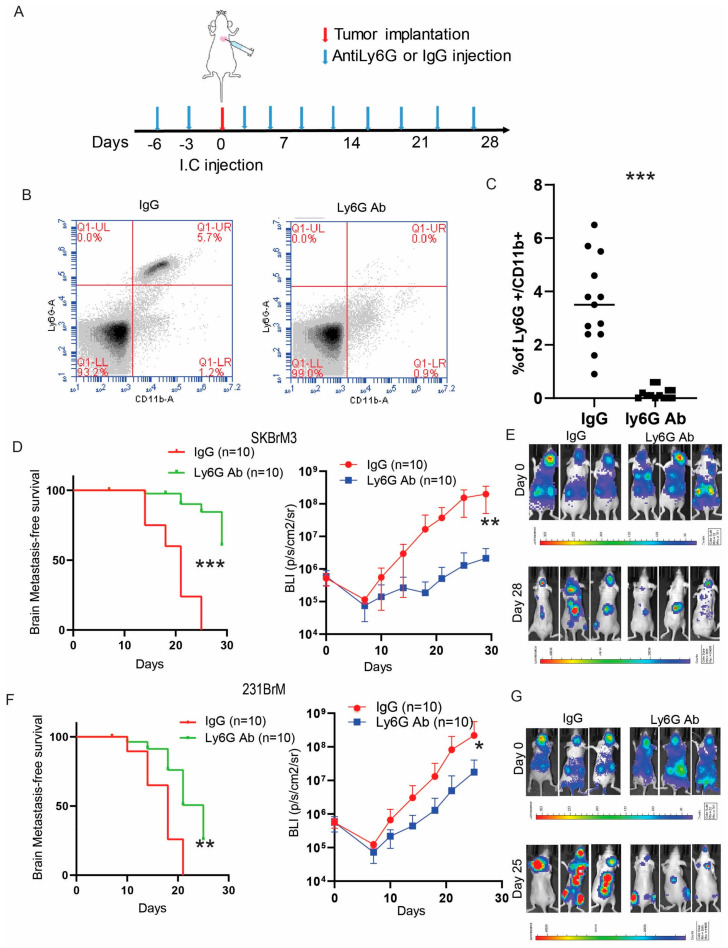
Depletion of neutrophils suppresses brain metastasis in vivo (**A**) Injection timeline and scheme. (**B**) PMBCs from mice treated with IgG or Ly6G antibodies were stained with neutrophil marker LY6G and CD11b antibodies and analyzed by flow cytometry. (**C**) Quantification of Figure B. Brain metastasis-free survival (Kaplan–Meier analysis) and brain bioluminescence of (**D**). SKBrM3 cells and (**F**) 231BrM cells after IC injection in mice treated with IgG or Ly6G antibody. (**E**,**G**) Representative picture of mice at day 0 and endpoint after IC injection (* *p*  <  0.05, ** *p*  <  0.01, or *** *p* < 0.001).

**Figure 5 cancers-15-02626-f005:**
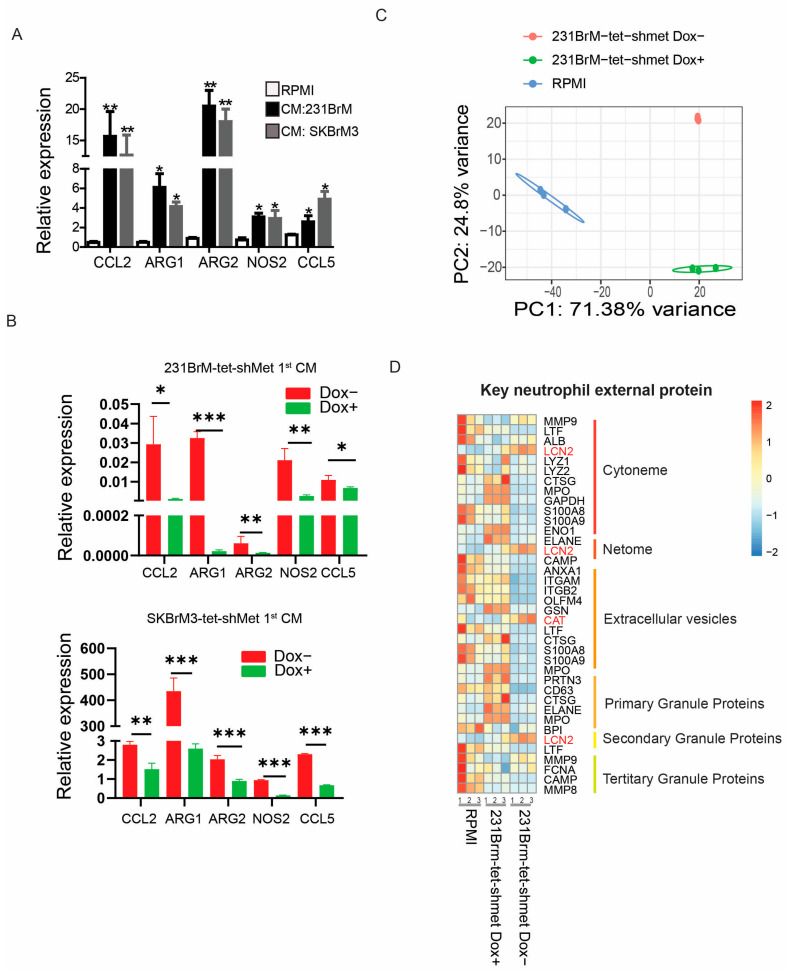
c-Met in cancer cells drives tumor-associated neutrophil remodeling (**A**) qPCR analyses of N2 neutrophil markers in mouse neutrophils treated with indicated CM. (**B**) N2 neutrophil markers were quantified by qPCR in mouse neutrophils treated with indicated CM (**C**) Principal Analysis of RNA sequencing data of neutrophils treated with indicated CM. (**D**) Heatmap of key neutrophils external proteins from RNA sequencing (* *p*  <  0.05, ** *p*  <  0.01, or *** *p* < 0.001).

**Figure 6 cancers-15-02626-f006:**
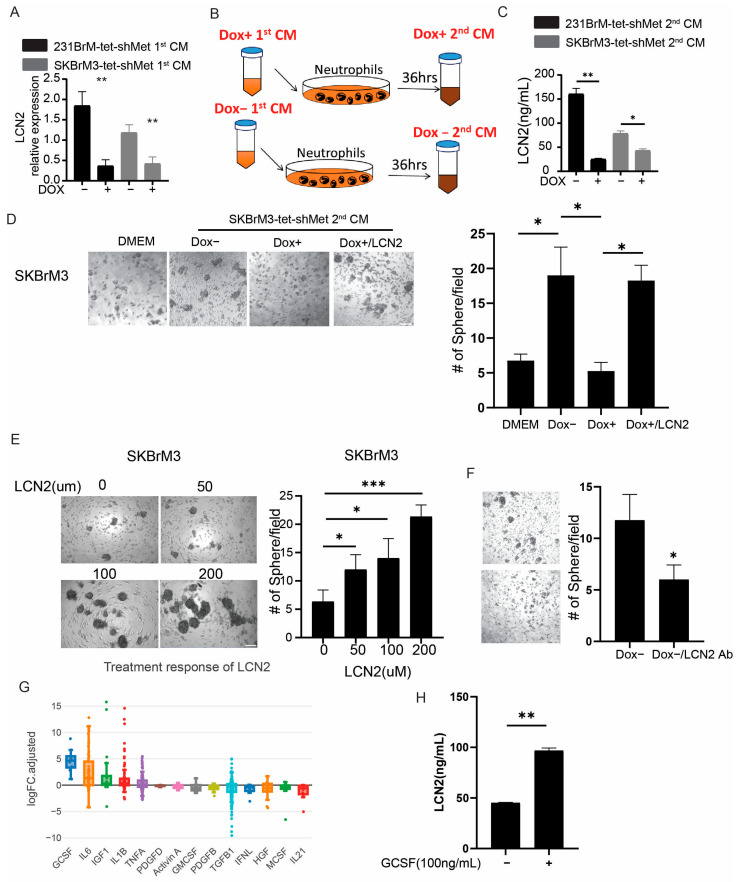
Neutrophils promote cancer stemness by LCN2 (**A**) LCN2 expression in mouse neutrophils was analyzed by qPCR after being treated by indicated CM or RPMI. (**B**) Scheme of 2nd CM preparation. (**C**) LCN2 level in 2nd CM was examined by ELISA. (**D**) Cancer cells were treated by indicated 2nd CM with or without LCN2 (100 µm) for 48 h, followed by mammary sphere formation assay. Left, representative picture at day 5. Scale bar: 100 µm. Right, quantification of spheres per well. (**E**) Cancer cells were cultured in media with indicated recombinant mouse LCN2 concentration for 48 h, followed by a mammary sphere formation assay. Scale bar: 100 µm. Left, representative picture at day 5. Right, quantification of spheres per well. (**F**) SKBrM3 cells were treated with Dox− 2nd CM and anti LCN2 antibody (100 µg/mL) followed by mammary sphere formation assay. (**G**) LCN2 expression after being treated by different cytokines from the Cytosig database. (**H**) LCN2 level in medium with or without G-CSF treatment was examined by ELISA (* *p*  <  0.05, ** *p*  <  0.01, or *** *p* < 0.001).

## Data Availability

The data that support the findings of this study are available from the corresponding author upon reasonable request.

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
