# Peer review of "c-Met Mediated Cytokine Network Promotes Brain Metastasis of Breast Cancer by Remodeling Neutrophil Activities"

_cancers, 2023, doi:10.3390/cancers15092626_

Round 1

Reviewer 1 Report

Brain metastasis is a lethal manifestation of cancer, and the mechanisms underlying its spread and progression remain poorly understood. Moreover, limited knowledge exists regarding the contributions of cancer cells and immune cells of the brain microenvironment to the metastatic process. Liu et al build upon their previous work, which described the c-MET pathway as a promoter of brain metastasis progression through its effect on the inflammatory brain microenvironment. In the current manuscript, the authors demonstrate how c-MET regulated cytokines influence metastatic progression to the brain. Specifically, they show that the c-Met pathway in brain metastatic cells enhances the expression of cytokines such as G-CSF, GM-CSF, CXCL1, which promote neutrophil migration and survival. They demonstrate that depletion of neutrophils suppresses brain metastasis in two mouse models of brain metastasis. The manuscript is well written, and the findings are interesting, however, there are a few issues related to the design and interpretation of the data that need to be addressed before publication.

1.       In Figure 3, the authors concluded that there is a reduction in neutrophil survival and migration in breast cancer cells treated with conditioned media from Dox-inducible c-MET knockdown. However, the reduction observed could be due to the inhibitory effect that Dox has on neutrophils. Therefore, a Dox-only control is necessary to arrive at the authors' conclusion. The same applies to Figure 6.

2.       In Figure 4C/E, the authors need to state the number of mice used per group, the p-value, and the statistical test used.

3.       The authors demonstrate the effect of LCN2 on primary mammosphere formation to conclude that it influences cancer stem cell properties. However, performing experiments with additional in vitro stem cell surrogate markers for breast cancer (e.g., CD44/CD24 or ALDH) and secondary mammosphere formation, a measure of the self-renewal of cancer cells, would strengthen the manuscript.

4.       The authors wrote, "These data suggest that neutrophil-derived LCN2 promotes stemness of brain metastatic cells." However, it is unclear what stemness of brain metastasis indicates, and whether it is established that cancer stem cells are equivalent to brain metastasis initiating cells.

5.       In Figure 6D, the authors demonstrate a decrease in mammosphere formation in DOX-treated cMET knockdown cells, but it is not clear if this reduction in spheres is attributed to suppression of LCN2 levels in the cMET knockdown cells. To evaluate the effect of mammospheres specifically on LCN2, the authors need to add an anti-LCN2 neutralizing antibody as a control.

6.       Adding a timeline and injection site schema for the in vivo experiments would be very helpful.

Author Response

  1. In Figure 3, the authors concluded that there is a reduction in neutrophil survival and migration in breast cancer cells treated with conditioned media from Dox-inducible c-MET knockdown. However, the reduction observed could be due to the inhibitory effect that Dox has on neutrophils. Therefore, a Dox-only control is necessary to arrive at the authors' conclusion. The same applies to Figure 6.

Response: We are sorry for the confusion. As it states in the methods, results and figure legend, cancer cells were pretreated with doxycycline for 48 hours to induce knockdown of c-Met and then we changed the medium to serum free medium for CM preparation. There is no doxycycline in conditioned medium. Therefore, the reduction effect is not caused by doxycycline.

  1. In Figure 4C/E, the authors need to state the number of mice used per group, the p-value, and the statistical test used.

Response: We added mice numbers. P value and statistical test were described in method.

  1. The authors demonstrate the effect of LCN2 on primary mammosphere formation to conclude that it influences cancer stem cell properties. However, performing experiments with additional in vitro stem cell surrogate markers for breast cancer (e.g., CD44/CD24 or ALDH) and secondary mammosphere formation, a measure of the self-renewal of cancer cells, would strengthen the manuscript.

Response: We didn’t perform the stem cell marker experiment mainly because sphere formation assay is considered as the gold standard assay of self-renew ability. Secondly, all of our brain metastatic cell lines have very high expression of CD44 and undetectable level of CD24. Secondary mammosphere was not performed due to all the sphere formation assay involves pretreatment of cancer cells in CM or medium with/without LCN2 for 48 hour. The pretreatment effect may not last long enough for secondary sphere formation.

  1. The authors wrote, "These data suggest that neutrophil-derived LCN2 promotes stemness of brain metastatic cells." However, it is unclear what stemness of brain metastasis indicates, and whether it is established that cancer stem cells are equivalent to brain metastasis initiating cells.

Response: We are sorry for the confusion. We are not implying the connection of cancer stemness and brain metastasis initiating capability here. Brain metastatic cells simply means the cell lines used in this manuscript, specifically 231BrM and SKBRM3.  We have changed cells to cell lines to clarify.

  1. In Figure 6D, the authors demonstrate a decrease in mammosphere formation in DOX-treated cMET knockdown cells, but it is not clear if this reduction in spheres is attributed to suppression of LCN2 levels in the cMET knockdown cells. To evaluate the effect of mammospheres specifically on LCN2, the authors need to add an anti-LCN2 neutralizing antibody as a control.

       Response: we have added the data according to reviewer’s suggestion in Fig 6F.

  1. Adding a timeline and injection site schema for the in vivo experiments would be very helpful.

      Response: We have added timeline and injection site in Fig. 4A.

Reviewer 2 Report

The manuscript by Yin Liu,et al used the induciable c-Met knowdown cells form 231BrM and SKBrM3 to study the role of c-Met on brain metastasis. They demonstrated that c-Met knowdown inhibits the expression of cytokines involved in the recruitment of neutrophil, including CXCL1, and thus inhibit neutrophil attraction and brain metastasis. Further, they found the condition medium of c-Met-high cells induced LCN secretion from neutrophil, which in turn promotes self-renewal of cancer stem cells. This study is interesting and provides useful information about the interplay between cancer cells and neutrophil in the microinvironment of brain mets.

Minor

(1) Fig.6, the authors stated that Neutrophil promotes cancer stemness by LCN2. The result is not convincing, the authors need do a recovery experiment to prove that the effect of neutriphils on the stemness of cancer cells is really LCN2 dependent.

Author Response

Minor

  • 6, the authors stated that Neutrophil promotes cancer stemness by LCN2. The result is not convincing, the authors need do a recovery experiment to prove that the effect of neutriphils on the stemness of cancer cells is really LCN2 dependent.

Response: we have added the data according to reviewer’s comment in Fig 6D and F.

Reviewer 3 Report

In the paper named”c-Met mediated cytokine network promotes brain metastasis of breast cancer by remodeling neutrophil activities” authors tray to

Identified the molecular and pathogenic mechanisms of how crosstalk between innate immune cells and tumor cells facilitate the tumor progression in the brain which provides novel therapeutic targets for treating brain metastasis. Moreover their results elucidated how c-Met mediated inflammatory cytokines affect neutrophil functions during different stages of breast cancer brain metastasis and how LCN2 from N2 neutrophils promote tumor progression which provide insights of developing potential therapeutic strategies by targeting c-Met signaling and N2 neutrophils.

Only minor points are required

1)     Nothing about the human cohort of breast cancer patients used in the paper

2)     In Material and methods in plasmid and reagents a brief description about the plasmids used and the cell lines generation may be included.

3)     In Material and methods in mammary sphere formation assay author say that they take the image after 7 days but they do not indicate if the cells medium is changed or not during this period

4)     In results section in paragraph between lines 198-200 author say that they perform an IPA analysis, however in material and methods this procedure is missing and none information about this analysis is found in the paper

5)     In line 211 they say that they stained clinical samples (n=26) derived from brain metastatic lesions with ELA2 antibody and observed the presence of neutrophils, but nothing is said about this samples, nothing about the clinical data.

6)     In figure 1C were author expose a flowchart of pathway analysis the cutoff parameters do not mach with those indicated in methods, please clarify. Regarding to this RNAseq author use c breast cancer cell lines however in methods only say that RNAseq was performed using neutrophils. Please clarify.

7)     In figure 2D and E how many samples author used per condition?

8)     How author determinate that the neutrophil are immature (line 273) using their morphology? The figure showing the results said in line 273 is missing. In this line author say that they studied nuclear morphology and expression of surface markers in Neutrophils.

9)     Related to question 3 in figure 6 author perform a sphere formation using a dose response of LCN2. This LCN2 is changed every day? Or only one dose is used?

Author Response

  • Nothing about the human cohort of breast cancer patients used in the paper

Response: We are sorry for the confusion. The cohort was briefly introduce in figure legend 2B. To further clarify, we have also added cohort analysis to material and method.

  • In Material and methods in plasmid and reagents a brief description about the plasmids used and the cell lines generation may be included.

Response: We have added the plasmid information.

3)     In Material and methods in mammary sphere formation assay author say that they take the image after 7 days but they do not indicate if the cells medium is changed or not during this period.

Response: No medium change were performed. There were only 1000 cells seeded in 500ul of medium. The medium is sufficient to support the cells for a long period.

4)     In results section in paragraph between lines 198-200 author say that they perform an IPA analysis, however in material and methods this procedure is missing and none information about this analysis is found in the paper.

      Response: We didn’t include IPA analysis in material and method as IPA analysis is a relatively simple analysis. Users only need to upload a gene list to generate the report. The analysis was briefly introduced in line 197-200. To further clarify, we added brief introduction in figure legend.

5)     In line 211 they say that they stained clinical samples (n=26) derived from brain metastatic lesions with ELA2 antibody and observed the presence of neutrophils, but nothing is said about this samples, nothing about the clinical data.

       Response: We are sorry for the missing of information. We have added Huamn Samples in material and methods.

6)     In figure 1C were author expose a flowchart of pathway analysis the cutoff parameters do not mach with those indicated in methods, please clarify. Regarding to this RNAseq author use c breast cancer cell lines however in methods only say that RNAseq was performed using neutrophils. Please clarify.

      Response: We are sorry for the confusion. The RNAseq describled in material and method refer to neutrophil RNA sequencing discussed in figure 5. The cuttoff parameters refer to the cut offs in volcano plot in sFig 3A. We have made edits to clarify. The cancer cell RNA sequencing was initially not included as it was done previously. However, we have added it to the material and method too.

7)     In figure 2D and E how many samples author used per condition?

      Response:. We have added mouse number to the figure.

8)     How author determinate that the neutrophil are immature (line 273) using their morphology? The figure showing the results said in line 273 is missing. In this line author say that they studied nuclear morphology and expression of surface markers in Neutrophils.

Response: During neutrophil maturation differentiation, the developing neutrophil changes its nuclear morphology from a round shape to a banded morphology into a segmented shape. Neutrophil nuclear shape and maturation stage are reviewed and illustrated in figure 1 of citation 11 (Coffelt, S.B.; Wellenstein, M.D.; de Visser, K.E. Neutrophils in cancer: neutral no more. Nat Rev Cancer 2016, 16, 431-446, doi:10.1038/nrc.2016.52.). For example, in the picture below, the cell on the left is a typical metamyelocyte and the cell on the right is a typical myelocyte. Both of them are immature neutrophils.

In line 273, we stated that neutrophils were isolated from bone marrow of C57/B6 mice and majority (~80%) of the isolated neutrophils are considered immature referring to the first picture and bar graph of sFig2. However, we made a mistake at labeling. RPMI should be untreated. We have made correction respectively.

9)     Related to question 3 in figure 6 author perform a sphere formation using a dose response of LCN2. This LCN2 is changed every day? Or only one dose is used?

Response: As it stated in line 375 -377 and 378-379, we pre-treated cancer cells with CM or LCN2 before sphere formation. LCN2 was not added to the sphere formation medium.

Round 2

Reviewer 1 Report

The authors have adequately addressed the criticisms. 

Author Response

Thanks